# Performance of In Silico Prediction Tools for the Detection of Germline Copy Number Variations in Cancer Predisposition Genes in 4208 Female Index Patients with Familial Breast and Ovarian Cancer

**DOI:** 10.3390/cancers13010118

**Published:** 2021-01-01

**Authors:** Louisa Lepkes, Mohamad Kayali, Britta Blümcke, Jonas Weber, Malwina Suszynska, Sandra Schmidt, Julika Borde, Katarzyna Klonowska, Barbara Wappenschmidt, Jan Hauke, Piotr Kozlowski, Rita K. Schmutzler, Eric Hahnen, Corinna Ernst

**Affiliations:** 1Center for Familial Breast and Ovarian Cancer, Center for Integrated Oncology (CIO), Medical Faculty, University Hospital Cologne, 50931 Cologne, Germany; louisa.lepkes@outlook.de (L.L.); mohamad.kayali@uk-koeln.de (M.K.); britta.bluemcke@uk-koeln.de (B.B.); jonas@drsweber.de (J.W.); sandra.schmidt__@uk-koeln.de (S.S.); julika.borde@uk-koeln.de (J.B.); barbara.wappenschmidt@uk-koeln.de (B.W.); jan.hauke@uk-koeln.de (J.H.); rita.schmutzler@uk-koeln.de (R.K.S.); eric.hahnen@uk-koeln.de (E.H.); 2Institute of Bioorganic Chemistry, Polish Academy of Sciences, 61-704 Poznan, Poland; msuszynska@ibch.poznan.pl (M.S.); kklonowska@bwh.harvard.edu (K.K.); kozlowp@ibch.poznan.pl (P.K.)

**Keywords:** breast/ovarian cancer susceptibility genes, HBOC, CNV, multigene panel sequencing

## Abstract

**Simple Summary:**

The identification of germline copy number variants (CNVs) by targeted nextgeneration sequencing frequently relies on in silico prediction tools with unknown sensitivities. We investigated the performances of four in silico CNV prediction tools in 17 cancer predisposition genes in a large series of 4208 female index patients with familial breast and/or ovarian cancer. We identified 77 CNVs in 76 out of 4208 patients; six CNVs were missed by at least one of the prediction tools. Experimental verification of in silico predicted CNVs is required due to high frequencies of false positive predictions. For female index patients with familial breast and/or ovarian cancer, CNV detection should not be restricted to *BRCA1/2* due to the relevant proportion of CNVs in further cancer predisposition genes.

**Abstract:**

The identification of germline copy number variants (CNVs) by targeted next-generation sequencing (NGS) frequently relies on in silico CNV prediction tools with unknown sensitivities. We investigated the performances of four in silico CNV prediction tools, including one commercial (Sophia Genetics DDM) and three non-commercial tools (ExomeDepth, GATK gCNV, panelcn.MOPS) in 17 cancer predisposition genes in 4208 female index patients with familial breast and/or ovarian cancer (BC/OC). CNV predictions were verified via multiplex ligation-dependent probe amplification. We identified 77 CNVs in 76 out of 4208 patients (1.81%); 33 CNVs were identified in genes other than *BRCA1/2*, mostly in *ATM*, *CHEK2*, and *RAD51C* and less frequently in *BARD1*, *MLH1*, *MSH2*, *PALB2*, *PMS2*, *RAD51D*, and *TP53*. The Sophia Genetics DDM software showed the highest sensitivity; six CNVs were missed by at least one of the non-commercial tools. The positive predictive values ranged from 5.9% (74/1249) for panelcn.MOPS to 79.1% (72/91) for ExomeDepth. Verification of in silico predicted CNVs is required due to high frequencies of false positive predictions, particularly affecting target regions at the extremes of the GC content or target length distributions. CNV detection should not be restricted to *BRCA1/2* due to the relevant proportion of CNVs in further BC/OC predisposition genes.

## 1. Introduction

Targeted next-generation sequencing (NGS) is an established tool for the detection of germline variants in cancer predisposition genes. While variants involving a few nucleotides, i.e., single-nucleotide variants (SNVs) and short insertion/deletion events (indels), can be detected with high accuracy, the identification of larger genomic rearrangements (copy number variants (CNVs)) remains challenging. To avoid laborious wet lab analyses for CNV detection such as array comparative genomic hybridization (aCGH) or multiplex ligation-dependent probe amplification (MLPA) [1,2,3,4,5] for all genes of interest, several publicly, as well as commercially available in silico tools have been developed to predict CNVs using targeted NGS data, which are now commonly used for CNV prescreening. However, several studies suggested that existing tools for CNV detection using targeted NGS data show limited accuracy and robustness [6,7,8,9,10]. In our study, we investigated the comparative performances of four in silico CNV prediction tools, including one commercial tool incorporated in the CE-IVD-marked Sophia Genetics DDM pipeline and three established publicly available tools, namely ExomeDepth [11], GATK gCNV [12], and panelcn.MOPS [13] in a large study sample of 4208 female index patients with familial breast and/or ovarian cancer (BC/OC). ExomeDepth uses a beta-binomial model to normalize for technical noise during the preparation of a reference sample set and finally uses a hidden Markov model for CNV calling [11]. Although originally developed for usage with exome data, ExomeDepth is applicable also to gene panel data and has already been employed in a variety of studies for that purpose [8,9,14,15]. GATK gCNV was released in January 2018 as a utility within GATK v4 and combines a negative-binomial factor analysis for read depth modeling and a hierarchical hidden Markov model for modeling of copy number states in a simultaneous training phase and is applicable to whole genome sequencing, exome sequencing, as well as gene panel data [12]. panelcn.MOPS was explicitly developed for usage with gene panel data. After the application of several quality filters at the sample and target level, panelcn.MOPS chooses a set of input samples for the construction of normalized reference read counts, based on read count correlations. Then, a Poisson mixture model is applied to each target region separately prior to the final integer copy number estimation [13].

In our study, we focused on 17 established cancer predisposition genes including *ATM* (MIM 607585), *BARD1* (MIM 601593), *BRCA1* (MIM 113705), *BRCA2* (MIM 600185), *BRIP1* (MIM 605882), *CDH1* (MIM 192090), *CHEK2* (MIM 604373), *MLH1* (MIM 120436), *MSH2* (MIM 609309), *MSH6* (MIM 600678), *PALB2* (MIM 610355), *PMS2* (MIM 600259), *PTEN* (MIM 601628), *RAD51C* (MIM 602774), *RAD51D* (MIM 602954), *STK11* (MIM 602216), and *TP53* (MIM 191170), for which MLPA assays were available for the verification of in silico predicted CNVs. The prevalence of CNVs in established BC/OC predisposition genes is poorly studied, and current data are either limited to *BRCA1/2* [16,17,18,19] only or based on small study samples for some non-*BRCA1/2* cancer predisposition genes [2,20,21].

## 2. Results

### 2.1. CNV Predictions Using the Sophia Genetics DDM

The Sophia Genetics DDM software predicts CNVs with high confidence or medium confidence and classifies the remaining target regions either as normal with high confidence or medium confidence or undetermined. In the study sample of 4208 female patients with familial BC/OC, the Sophia Genetics DDM software predicted 134 CNVs, of which 103 were classified as CNVs with high confidence and 31 were classified as CNVs with medium confidence. Of the 134 predicted CNVs, seventy-seven (57.46%) could be verified by MLPA, and the remaining 57 (42.54%) could not be verified (Table 1). All verified CNVs are listed in Appendix A. False positive predictions predominantly affected target region CHEK2_ex07 (n = 23). In addition, the Sophia Genetics DDM software classified 257 target regions as normal with medium confidence and another 152 target regions as undetermined. Several target regions were prone to be classified as normal with medium confidence or undetermined, predominantly CHEK2_ex07 (n = 94), BRCA2_ex13 (n = 33), CHEK2_ex05 (n = 29), MLH1_ex15 (n = 19), BRCA1_ex09 (n = 11), and BRCA2_ex12 (n = 11). MLPA analyses of all target regions that were classified as normal with medium confidence or undetermined revealed no CNV, and the MLPA test result for one sample was not evaluable.

### 2.2. CNV Predictions Using ExomeDepth, GATK gCNV, and panelcn.MOPS

ExomeDepth predicted 91 CNVs in 90 samples. Of these 91 CNVs, seventy-two were identical to true positive CNV predictions using the Sophia Genetics DDM software. Another seven CNV predictions were identical to false positive CNV predictions by the Sophia Genetics DDM software, most of which affect target region CHEK2_ex07 (n = 5). The remaining 12 CNV predictions, again mostly affecting CHEK2_ex07 (n = 6), could not be verified by MLPA. A total of five true positive CNVs were missed by ExomeDepth (Table 2).

GATK gCNV predicted 370 CNVs in 305 samples. Of those, seventy-two were identical to true positive CNVs, and another two were identical to false positive CNV predictions of the Sophia Genetics DDM Software. The majority (98%) of the remaining 296 GATK gCNV predictions corresponded to one of the following target regions: BRCA2_ex11 (n = 175), PTEN_ex02 (n = 61), or BRCA1_ex10 (n = 55), suggesting false positive calls. Thus, we refrained from MLPA verification of CNV predictions affecting these three target regions if GATK gCNV was the only predicting tool. The remaining six predicted CNVs could not be verified by MLPA. A total of five true positive CNVs were missed by GATK gCNV (Table 2).

For panelcn.MOPS, nine samples did not pass the sample quality filter and therefore were excluded from further analyses. In the remaining 4199 samples, panelcn.MOPS predicted the highest number of 1254 CNVs in 727 samples. Of those, seventy-four were identical to true positive CNVs by the Sophia Genetics DDM software and another 39 were identical to false positive Sophia Genetics DDM software predictions, predominantly affecting CHEK2_ex07 (n = 20). Three quarters of the remaining 1141 predicted CNVs accumulated in four target regions, i.e., CHEK2_ex07 (n = 429), BRCA2_ex13 (n = 179), BRCA1_ex08 (n = 164), and CHEK2_ex05 (n = 80). For panelcn.MOPS, we restricted MLPA verification to suspicious target regions that were affected in less than 10% of all samples per sequencing run (data not shown). None of the resulting 228 MLPA analyses verified a predicted CNV. A total of three true positive CNVs were missed by panelcn.MOPS (Table 2).

### 2.3. CNV Landscape in the Study Sample

In total, seventy-seven true positive CNVs were identified in 76 patients, with one individual carrying two deletions in different genes, namely in *BRCA2* and *PMS2* (Appendix A). This leads to an overall CNV prevalence of 1.81% in the study patients (76/4208). CNV prevalence was highest in index patients with both BC and OC (3.17%, 4/126), second-highest in patients with BC (1.81%, 66/3639), and lowest in OC patients (1.35%, 6/443). Most CNVs were present in the *BRCA1* (29 deletions, 9 duplications), *CHEK2* (12 deletions, 0 duplications), *BRCA2* (3 deletions, 3 duplications), and *ATM* genes (1 deletion, 5 duplications) (Figure 1). CNVs in the *BARD1*, *MLH1*, *MSH2*, *PALB2*, *PMS2*, *RAD51C*, *RAD51D*, and *TP53* genes, respectively, were less prevalent. No CNVs were identified in the *BRIP1*, *CDH1*, *MSH6*, *PTEN*, and *STK11* genes. Among the 77 true positive CNVs, several were recurrent (11× CHEK2_ex09 to CHEK2_ex10; 7× BRCA1_ex01 to BRCA1_ex02; 7× BRCA1_ex16 only; 4× ATM_ex62 to ATM_ex63; 3× RAD51C_ex05 to RAD51C_ex09; 2× BRCA1_ex12; 2× BRCA1_ex21; 2× BRCA1_ex22, Figure 2). Deletion of exon 9 to exon 10 in the *CHEK2* gene has already been described as significantly associated with an increased risk for the development of BC [20]. Deletions covering exon 5 in *RAD51C* have also been repeatedly characterized and described as pathogenic [21,22]. However, the clinical relevance of some CNVs in non-*BRCA1/2* genes is still unclear, although they may be observed relatively frequently, e.g., the duplication of exon 62 to exon 63 in *ATM* is reported as a variant of uncertain significance in the ClinVar database [23] (ClinVar Variation ID 429035.1) as of 14 December 2020.

### 2.4. Determinants of Sophia Genetics DDM Predictions with Reduced Confidence

All in silico CNV prediction tools employed in this study use read depth-based approaches, i.e., CNV calling is based on the hypothesis that a CNV determines the relative read depth per target region. Thus, low or highly fluctuating read depths of a target region likely complicate accurate CNV prediction. We suggest that target region sequencing coverage along with target region characteristics, such as GC content, length, and mappability, determined the accumulation of false positive CNV predictions (n = 57) and of classifications of target regions as either normal with medium confidence (n = 274) or undetermined (n = 131) by the Sophia Genetics DDM software. Indeed, the 462 undetermined, normal with medium confidence, or false positive predictions were associated with low sequencing coverage of corresponding target regions (Spearman’s rank correlation coefficient $\rho =-0.33,p=4.35\times 10^{-8}$, Figure 3). Target-specific values of GC content, length, and mappability were ascertained using CODEX2 [24] (see Section 4.6). The highest absolute Spearman’s rank correlation coefficients were observed for the deviation of the GC content from 0.5, suggesting that extreme GC values complicate reliable CNV prediction (Figure 3). Furthermore, significant correlations were observed for target region lengths (*p* values $<10^{-3}$), i.e., CNV prediction was complicated by short target region lengths (Figure 3). A significant impact of reduced mappability on the reliability of CNV predictions could not be observed (Figure 3). Several target regions, such as CHEK2_ex07, BRCA2_ex13, or CHEK2_ex05, appear to be especially challenging for Sophia Genetics DDM and also ExomeDepth or panelcn.MOPS. Concordant with our findings, all three target regions were characterized by a low GC content below 30%, with a mean GC content of 40% (range 24–73%) observed for all target regions (Appendix A). In addition, all three target regions were shorter than 0.25 kbp (Figure 3), with a mean target region length of 0.32 kbp (range 0.18–5.04 kbp) observed for all target regions. Frequent and thus likely false positive calls of BRCA2_ex11 and BRCA1_ex10 were unique to GATK gCNV. These target regions were the two largest among all target regions with 5.04 kbp and 3.48 kbp, respectively.

## 3. Discussion

The Sophia Genetics DDM software showed the highest sensitivity in our analyses and outperformed ExomeDepth, GATK gCNV, and panelcn.MOPS at default parameter settings. A total of six true positive CNVs were missed by at least one of the latter three tools (Table 1). Combining all three publicly available CNV prediction tools used in this investigation only marginally increased the overall sensitivity, as three CNVs were missed by all tools (Table 2). However, the performance of the three publicly available CNV prediction tools may improve when using settings other than the default. Under default parameter settings, none of the publicly available CNV prediction tools detected a true positive CNV missed by the Sophia Genetics DDM software. For the Sophia Genetics DDM software, the probability of a CNV prediction to represent a true positive CNV, i.e., its positive predictive value (PPV), was 57.46% (77/134). For ExomeDepth, the PPV was 79.12% (72/91), though with reduced sensitivity in comparison to Sophia Genetics DDM. For GATK gCNV and panelcn.MOPS, the obtained values of PPV were only 19.46% (72/370) and 5.90% (74/1254), respectively, and also associated with reduced sensitivity compared with Sophia Genetics DDM. In a research setting in which a reduced sensitivity may be tolerable, we suggest that ExomeDepth is superior to GATK gCNV and panelcn.MOPS under the default parameter settings due to the lower proportion of false positive CNV predictions. In addition, our investigation may be useful for the optimization of diagnostic gene panel design, which is still the most prevalent NGS method used for the identification of likely pathogenic germline mutations in BC/OC predisposition genes. Target region characteristics at the extremes of the target length or GC content distributions were likely affected by false positive CNV predictions and predictions with low confidence. We suggest that target region definitions may be optimized towards average target region length and average GC content in the overall gene panel. Regarding the CNV landscape observed in our study sample, forty-four of the 4208 patients carry *BRCA1/2* CNVs (1.05%). This CNV prevalence is somewhat lower than described by Myriad Genetics for a high-risk study sample with BC diagnosed under age 50 years, or ovarian cancer, or male breast cancer, in conjunction with two or more relatives similarly affected [16]. In that study, three-hundred five *BRCA1/2* CNVs were identified in 13,945 patients of European descent (2.19%). In 16,615 patients of European descent who did not meet these high-risk criteria, however, the *BRCA1/2* CNV prevalence was 0.39% (64/16,615) [16]. Thus, the *BRCA1/2* CNV prevalence observed in our study lies within the expected range and may reflect the stringency of our inclusion criteria used. Besides *BRCA1/2*, a significant proportion of CNVs was identified in non-*BRCA1/2* genes, affecting 33 of the 4208 patients (0.76%), mostly in the *ATM*, *CHEK2*, and *RAD51C* cancer predisposition genes and less frequently in *BARD1*, *MLH1*, *MSH2*, *PALB2*, *PMS2*, *RAD51D*, and *TP53*.

This study has limitations. The absolute sensitivity of Sophia Genetics DDM remains unknown. For assessing this quality parameter in our study, MLPA analyses of 17 genes in 4208 samples would have been required, which is beyond the scope of this investigation. We also refrained from characterizing verified CNVs in more detail by identification of break point positions, such as by aCGH analysis. Furthermore, we have to point out that our results refer to blood-derived DNA samples and are not transferable to analyses of tumor DNA and DNA samples derived from formalin-fixed and paraffin-embedded (FFPE) tissues.

## 4. Materials and Methods

### 4.1. Study Sample

A total of 4208 consecutive female index patients with familial BC/OC were included. Of those, three-thousand six-hundred thirty-nine patients were affected by BC, 443 patients by OC, and 126 patients by BC and OC. All index patients met the inclusion criteria of the German Consortium for Hereditary Breast and Ovarian Cancer (GC-HBOC) for germline testing [25]. Written informed consent was obtained from all patients, and ethical approval was granted by the ethics committee of the University of Cologne (07-048).

### 4.2. Targeted Next-Generation Sequencing

Genomic DNA was isolated from venous blood samples using standard methods. Targeted NGS was performed using a customized hybrid capture gene panel (TruRisk^®^v1 gene panel, Agilent SureSelect, QXT protocol) on an Illumina NextSeq500 sequencing device (San Diego, IL, USA). The TruRisk^®^v1 gene panel covers 34 (candidate) cancer predisposition genes. NGS analyses were performed in a routine diagnostic setting at the Center for Familial Breast and Ovarian Cancer, Cologne, Germany, between November 2015 and October 2017. Targeted NGS analyses of DNA samples derived from 4208 female patients included in this investigation were distributed over 111 sequencing runs, with run sizes varying from 30 to 96 samples (mean = 44.72, median = 46).

### 4.3. Selection of Target Regions

We considered protein-coding exons of 17 cancer predisposition genes according to the canonical hg19 RefSeq transcripts of *ATM* (NM_000051.3), *BARD1* (NM_000465.3), *BRCA1* (NM_007294.3), *BRCA2* (NM_000059.3), *BRIP1* (NM_032043.2), *CDH1* (NM_004360.4), *CHEK2* (NM_007194.3), *MLH1* (NM_000249.3), *MSH2* (NM_000251.2), *MSH6* (NM_000179.2), *PALB2*  (NM_024675.3), *PMS2* (NM_000535.6), *PTEN* (NM_000314.6), *RAD51C* (NM_058216.2), *RAD51D* (NM_002878.3), *STK11* (NM_000455.4), and *TP53* (NM_ 000546.5). For *BRCA1*, we additionally included the non-coding exon 1. We excluded the exons 12 to 15 of the *PMS2* gene and the exon 3 of the *PTEN* gene due to highly homologous regions in pseudogenes. CNVs are generally defined as gains or losses of at least 50 bp of genomic DNA [26]. In this investigation, we focused on CNVs that span at least one target region. Most predefined target regions span one exon, while several predefined target regions span two or three exons when located in very close proximity (*ATM* exons 2 and 3, exons 19 and 20, exons 21 and 22, and exons 41 and 42; *BRCA2* exons 5 and 6; *MSH6* exons 9 and 10; *PALB2* exons 2 and 3; *RAD51D* exons 7 and 8; *STK11* exons 4 and 5; *TP53* exons 2 to 4 and exons 8 and 9). This resulted in a total of 274 selected target regions in 17 genes. The TruRisk^®^v1 34 gene panel overall contains 571 target regions. Following NGS, the average target region coverage for all 571 target regions ranged from 623 to 4643, with a mean of 2471.

### 4.4. In silico Prediction of Germline Copy Number Variations in Cancer Predisposition Genes

For CNV prediction, we employed the commercial Sophia Genetics DDM pipeline v3.4.0–4.6.2 (Sophia Genetics, Saint-Sulpice, Switzerland) and three publicly available in silico CNV prediction tools, namely ExomeDepth [11], GATK gCNV [12], and panelcn.MOPS [13]. Samples rejected for quality reasons by the Sophia Genetics DDM software were not included in this investigation. All analyses were performed using all samples per run and all 571 TruRisk^®^v1 target regions, starting from unmapped sequencing reads in the FASTQ format.

For CNV prediction with ExomeDepth, GATK gCNV [12], and panelcn.MOPS, sequence reads were mapped to the human reference genome assembly GRCh37 including decoy sequences (hs37d5) using BWA-MEM of Burrows-Wheeler Aligner v0.7.15 [27,28] and processed according to the GATK BestPractices, including duplicate marking, realignment of insertions and deletions, and quality recalibration using GATK v3.8 [29,30].

Input data for ExomeDepth [11], i.e., read counts per sample and target region, were received via the built-in method getBamCounts(), and reference samples were selected using select.reference.set() under the specification of individual target lengths (argument bin.length). Calling function CallCNVs() was run with default arguments, and in-house scripts were employed to extract CNV calls from the resulting CSV files. CNV calls from all samples under consideration were reported, irrespective of correlations between reference and test counts below 0.97, which point towards low quality samples due to the publishers. All analyses were run with ExomeDepth v1.1.10 under R v3.6.2.

Input HDF5 files for GATK gCNV were generated using GATK’s CollectReadCounts utility with the interval merging rule set to OVERLAPPING_ONLY. DetermineGermlineContig Ploidy and GermlineCNVCaller were run in COHORT mode under the specification of 0.97 for the prior probability of ploidy state 2 and 0.01 of ploidy states 0, 1, and 3 for all chromosomes. All analysis steps including the generation of the output VCF files via PostprocessGermline CNVCalls were run with GATK v4.1.0.0.

Input data for panelcn.MOPS, i.e., read counts per sample and target region, were received by employing the multicov utility of bedtools v2.26 [31]. For CNV detection, functions panelcn.mops() and integerCopyNumber() of panelcn.MOPS v1.6.0 were run with default parameters under R v3.6.2. Samples that were classified as Bad test samplewere treated as sample outliers and not considered for final CNV output.

### 4.5. Multiplex Ligation-Dependent Probe Amplification Analyses

Verification of predicted CNVs in selected cancer predisposition genes was performed by MLPA analyses using SALSA^®^ MLPA^®^ Probemixes (MRC-Holland, Amsterdam, The Netherlands) according to the manufacturer’s protocol: P041 (*ATM*), P042 (*ATM*), P002 (*BRCA1*), P087 (*BRCA1*), P045 (*BRCA2*/*CHEK2*), P077 (*BRCA2*), P240 (*BRIP1*), P083 (*CDH1*), P190 (*CHEK2*), P003 (*MLH1*/*MSH2*), P248 (*MLH1*/*MSH2*), P072 (*MSH6*), P260 (*PALB2*/*RAD50*/*RAD51C*/*RAD51D*), P008 (*PMS2*), P105 (*PTEN*), P101 (*STK11*), and P056 (*TP53*). Data were analyzed using the Coffalyzer.Net software v140429.1058 (MRC-Holland). For the verification of CNVs in the *BARD1* gene, we used a non-commercial MLPA assay previously developed for scientific purposes [32]. The first coding exons of the *MSH2* and *RAD51D* genes, as well as exon 16 of the *CHEK2* gene were not covered by any of the MLPA kits. Thus, these three exons were excluded from further analysis, resulting in a final set of 271 selected target regions (Appendix A). A summarizing overview of our MLPA testing strategy and the MLPA analyses performed is given in Appendix A.

### 4.6. Extraction of Sequencing Target Characteristics

Estimates of averaged read abundance per target regions $\bar{RC}$ were obtained by extracting counts of mapped reads per sample and target using the multicov utility of bedtools v2.26 [31], averaging the resulting values run-wise and subsequently over all runs and final normalization by target length in base pairs to account for an expected linear relationship between target length and read counts.

Mappability is a measure of a reference sequence’s capability to produce reads that map uniquely to a corresponding reference genome. Therefore, mappability depends on genomic sequence characteristics, read lengths, the reference genome, and the amount of allowed mismatches. Mappability takes values between 0 and 1, whereby 1 refers to genomic regions that are expected to produce uniquely mapping reads exclusively, and a region with a mappability of 0 is expected to produce no uniquely mapping reads at all. Several approaches for the computation of mappability exist, ranging from simple investigation of sequence motifs to sophisticated simulation approaches, resulting in ambiguous formal definitions. Here, the getmapp() utility of CODEX2 v1.3 [24] was employed for the determination of mappability values, using pre-computed results, which are based on the construction of consecutive 90 bp reads and under permission of 2 mismatches [33]. GC content per target region were received employing the getgc() utility of CODEX2 [24].

## 5. Conclusions

All four in silico CNV prediction tools show notable amounts of false positive predictions, and therefore, verification of predicted CNVs by either aCGH [2,3], MLPA [1,5], long-read sequencing [10,34], or other methods is required. False positive predictions and predictions with low confidence accumulate for target regions with extreme target lengths and/or GC contents. Therefore, our findings should be taken into account for the optimization of diagnostic gene panel design. We suggest that target region definitions may be optimized towards average target region length and average GC content in the overall gene panel. In the framework of genetic counseling for persons at risk for familial BC/OC, CNV detection should be included in routine germline diagnostics for all BC/OC predisposition genes and may not be restricted to *BRCA1/2*, as a relevant proportion of women in our study sample (0.76%) were affected by CNVs in non-*BRCA1/2* genes.

## Figures and Tables

**Figure 1 cancers-13-00118-f001:**
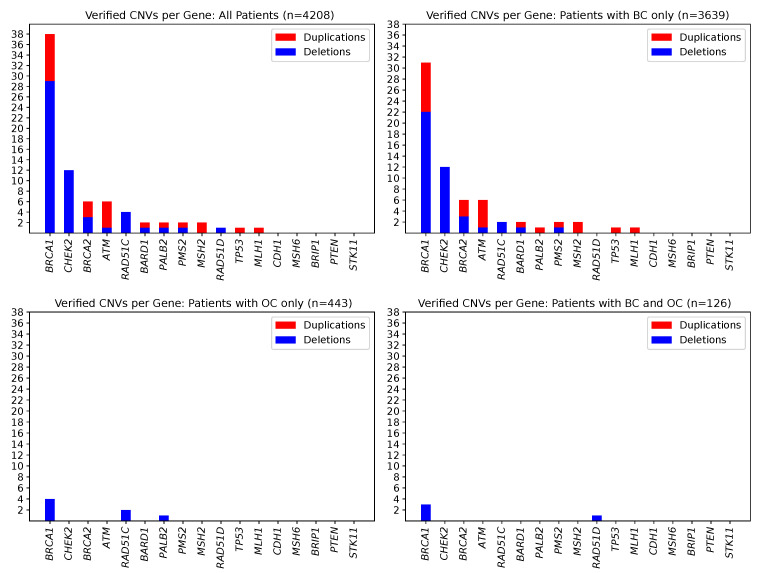
Counts of observed and MLPA-verified CNVs in 17 cancer predispositions genes. Deletions are shown in blue, and duplications are shown in red. Seventy-seven CNVs could be confirmed affecting 76 patients. (Upper left) Overall study sample of 4208 individuals. (Upper right) For the subgroup of 3639 individuals affected by BC. (Lower left) For the subgroup of 443 individuals affected by OC. (Lower right) For the subgroup of 126 individuals affected by BC/OC.

**Figure 2 cancers-13-00118-f002:**
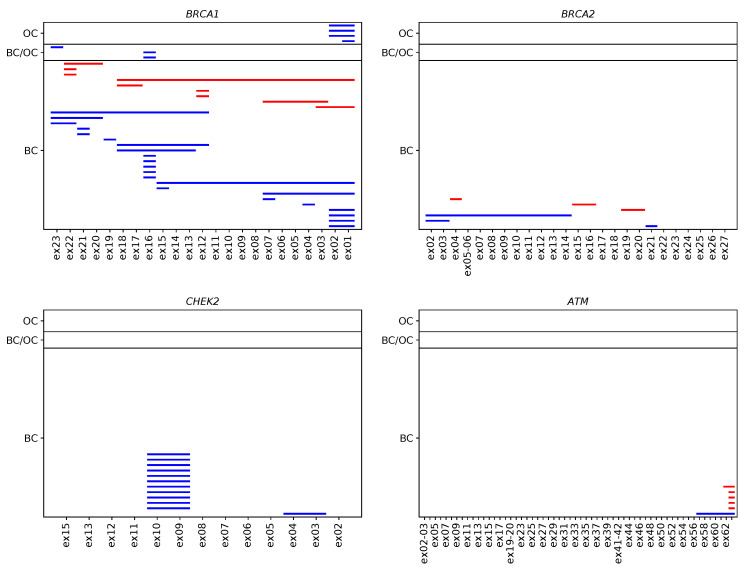
Verified CNVs in the four most frequently affected BC/OC predisposition genes in 4208 individuals according to phenotype. Each line represents a CNV spanning the corresponding sequencing targets in *BRCA1*, *BRCA2*, *CHEK2*, and *ATM*. Deletions are shown in blue, and duplications are shown in red.

**Figure 3 cancers-13-00118-f003:**
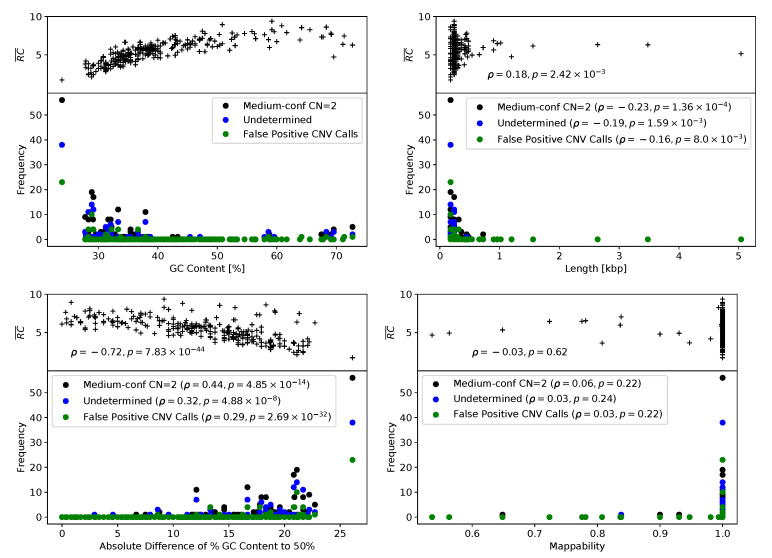
Read abundances and Sophia Genetics DDM outcome depending on target region characteristics. Averaged read abundance per target region ($\bar{RC}$; see Section 4.6) and counts of medium-confidence normal copy number, undetermined, and false positive predictions of the Sophia Genetics DDM Software dependent on (upper left) GC content, (lower left) absolute difference of GC ratio to 50%, (upper right) target length, and (lower right) mappability.

**Table 1 cancers-13-00118-t001:** Results of the Sophia Genetics DDM CNV predictions according to the confidence level provided by the Sophia Genetics DDM software, type of CNV, and the number of adjacent target regions affected. CNV predictions were considered as predictions with high confidence if at least one of the included target regions was predicted with high confidence. PPV = positive predictive value.

	Predicted CNVs	True Positive CNVs	PPV (%)
Overall	134	77	57.46
High confidence	103	75	72.82
Medium confidence	31	2	6.45
Deletions	63	53	84.13
Duplications	71	24	33.80
1 target region	70	25	35.71
>1 target region	64	52	81.25

**Table 2 cancers-13-00118-t002:** True positive CNVs predicted by Sophia Genetics DDM, but missed by either ExomeDepth (ED), GATK gCNV (GATK), or panelcn.MOPS (pcnMOPS). yes = CNV was predicted by the respective tool. no = CNV was missed by the respective tool.

Sample	CNV Type	Start	Stop	Target Regions	ED	GATK	pcnMOPS
44–22	deletion	BARD1_ex04	BARD1_ex01	4	yes	no	yes
14–15	duplication	BRCA1_ex22	BRCA1 ex22	1	no	no	no
89–01	duplication	BRCA2_ex04	BRCA2_ex04	1	no	no	no
9–25	duplication	BRCA2_ex19	BRCA2_ex20	2	no	yes	yes
49–28	deletion	BRCA2_ex02	BRCA2_ex14	12	yes	yes	no
29–19	duplication	PALB2_ex11	PALB2_ex11	1	no	no	no
16–27	duplication	TP53_ex08-09	TP53_ex02-04	5	no	no	yes

## Data Availability

The data presented in this study are available on request from the corresponding author. The data are not publicly available due to privacy.

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
