# Peer review of "Performance of In Silico Prediction Tools for the Detection of Germline Copy Number Variations in Cancer Predisposition Genes in 4208 Female Index Patients with Familial Breast and Ovarian Cancer"

_cancers, 2021, doi:10.3390/cancers13010118_

Round 1

Reviewer 1 Report

The paper describes the comparisons of four different methods for the detection of CNV in sugh driver cancer genes, most of them involved in genome stability. The topic is hot, although an attempt to better define the features of homologous recombination deficiency is still challenging. 

Nevertheless, the paper shows as a commercial DDM tool is outperforming in its capability to call CNV as compared to the non-commercial ones. It would have been very interesting to have at list one more commercial kit to compare, since the algorithms  used can influence the rate of detection (please, see what has been reviewed in Expert Rev Mol Diagn 2019 Sep;19(9):795-802. The definitive confirmation of CNV is generally performed by MLPA or MAQ, when possible, since both methods can be complimetary, particularly in presence of discrepancy.
PArticularly for this large cohort of patients investigate, authors should clarify if the MLPA analysis was performed on fresh or biobanked DNA: this is important, since MLPA performs very well on fresh DNA. This detail is missing. In addition, authors should indicate why they did not further evaluate by cGH array those few samples with true positive CNV in order to better indicate the break points.   These limits should be underlined in the conclusions.

Reviewer 2 Report

The study tested the performances of four in silico germline copy number variants (CNV) prediction tools. Authors compared performance of one commercial and three non-commercial tools using 17 cancer predisposition genes in 4,208 female patients with familial breast and/or ovarian cancer (BC/OC). The study reports low performance number (77 CNVs in 76 out of 4,208 patients (1.81%)) using the selected gene panel. However, 33 CNVs were identified in genes other than BRCA1/2, mostly in ATM, CHEK2, and RAD51C. The main conclusion indicates that CNV detection should not be restricted to BRCA1/2. Importantly, the study reports false positive outcomes for several tools/kits , thus, indicating that those approaches should be used cautiously.

The study is useful in BC/OC diagnostics and well-written.

 However, there are several points to address.

  1. The study should present a scheme/graph that demonstrates the testing protocol with clear links from a testing tool to the tested gene panel and the outcome of the test; it will allow to visualize the productivity of the test and the usefulness of the tool for diagnostics.
  2. Discussion section should be extended and describe the new target CNVs detected in this study; what is found previously for these “new” CNVs? Why they can be better than BRCA1/2?
  3. Conclusion section is awkward/obscure and should be presented clearly.

Reviewer 3 Report

In this study the authors investigate the comparative performances of 4 in silico CNV prediction tools, including one commercial (Sophia Genetics DDM) and 3 non-commercial (ExomeDepth, GATK gCNV, and panel.cnMOPS) in a large study with 4208 patients with familiar breast or ovarian cancer. They conclude that the Sophia genetics DDM software is the most sensitive and outperforms the other software tested. In light of their results, they underline also the fact that CNV detection should not be restricted only to BRCA1/2 due to the relevant proportion of CNVs in further breast/ovarian cancer predisposition genes.

The paper is well written, the results are clearly explained and discussed. Importantly, there are comparisons also with other methods of CNV’s investigations (i.e. the Myrian Genetics method).
